**Anthropogenic Fine Particulate Matter Pollution Will Be Exacerbated in Eastern**
**China Due to 21st-Century GHG Warming**
Huopo Chen[1, 2*], Huijun Wang[2, 1], Jianqi Sun[1, 2], Yangyang Xu[3], and Zhicong Yin[2]
[1] *Nansen-Zhu International Research Centre, Institute of Atmospheric Physics,*
*Chinese Academy of Sciences, Beijing, China*
[2] *Collaborative Innovation Center on Forecast and Evaluation of Meteorological*
*Disasters, Nanjing University for Information Science and Technology, Nanjing,*
*China*
[3] *Department of Atmospheric Sciences, Texas A&M University, College Station Texas,*
*USA*
**Corresponding author:** Huopo Chen (chenhuopo@mail.iap.ac.cn)
**Address:** Nansen-Zhu International Research Centre, Institute of Atmospheric
Physics, Chinese Academy of Sciences, PO Box 9804, Beijing 100029,
China
**Email:** chenhuopo@mail.iap.ac.cn
**Tel:** (+86)010-82995057

## Abstract

China has experienced a substantial increase in severe haze events over the past several decades, which is primarily attributed to the increased pollutant emissions caused by its rapid economic development. The climate changes observed under the warming scenarios, especially those induced by increases in greenhouse gases (GHG), are also conducive to the increase in air pollution. However, how the air pollution changes in response to the GHG warming has not been thoroughly elucidated to date. We investigate this change using the century-long large ensemble simulations with the Community Earth System Model 1 (CESM1) with the fixed anthropogenic emissions at the year 2005. Our results show that although the aerosol emission is assumed to be a constant throughout the experiment, anthropogenic air pollution presents positive responses to the GHG-induced warming. The anthropogenic $PM_{2.5}$ concentration is estimated to increase averaged over eastern China at the end of this century, but varying from regions, with an increase over northwestern part of eastern China and a decrease over southeastern part. Similar changes can be observed for the light air pollution days. However, the severe air pollution days is reported to increase across eastern China at the end of this century, particularly around the Jing-Jin-Ji region. Further research indicates that the increased stagnation days and the decreased light precipitation days are the possible causes of the increase in $PM_{2.5}$ concentration, as well as the anthropogenic air pollution days. Estimation shows that the effect of climate change induced by the GHG warming can account for 11%-28% of the changes in anthropogenic air pollution days over eastern China. Therefore, in the

future, more stringent regulations on regional air pollution emissions are needed to

balance the effect from climate change.

## 1. Introduction

The extraordinarily rapid development of China has caused extremely high aerosol loading and gaseous pollutant emissions that have enveloped most regions across China in the recent decades. The increased pollutant emissions, particularly for the particulate matter finer than 2.5 μm in aerodynamic diameter ($PM_{2.5}$), generally result in severe haze events and present a major threat to public health (Gao et al., 2017; Tang et al., 2017; Wang, 2018), crop production (Tie et al., 2016), and regional climates (Cao et al., 2016). For example, the annual averaged $PM_{2.5}$ in Beijing exceeded 75 μg/m$^3$ during 2009-2016 (Fig. 1b), which more than three times the recommended 24-hour standard (25 μg/m$^3$) of the World Health Organization (WHO). This degeneration of the air pollution across China, which is similar to that in Beijing, is primarily caused by the integrated effects of high emissions and poor ventilation (Chen and Wang, 2015; Zhang et al., 2016a). Many efforts are thus underway to reduce emissions that cause severe haze pollutions. However, the question remains of whether climate change will offset or facilitate these efforts.

Recent studies have documented that the exacerbation of air quality over eastern China was partly modulated by meteorological conditions and climate variability that are generally conducive to the severe haze occurrences (Li et al., 2018; Liao and Chang, 2014; Wang and Chen, 2016; Yang et al., 2016; Zhang et al., 2014; Zhang et al., 2016b). Specifically, Wang *et al.* (2015) revealed that the shrinking Arctic sea ice favors less cyclone activity and a more stable atmosphere conducive to haze formation, which can explain approximately 45%-67% of the interannual to

interdecadal variability of winter haze days over eastern China. Besides Arctic sea ice,
other decadal variability and changes, including weak East Asian winter monsoon
(Jeong et al., 2017; Li et al., 2016; Yin et al., 2015), strong El Niño-Southern
oscillation (Gao and Li, 2015; Zhao et al., 2018), high Pacific decadal oscillation
(Zhao et al., 2016), and high Arctic oscillation (Cai et al., 2017), may have contributed.
In addition, the increasing winter haze days over eastern China may also be linked to
the low boundary layer height (Huang et al., 2018; Wang et al., 2018), weakened
northerly winds (Yang et al., 2017a), decreased relative humidity (Ding and Liu,
2014), and increased sea surface temperature (Xiao et al., 2015; Yin and Wang, 2016;
Yin et al., 2017).

Global warming generally presents an adverse impact on the haze pollution

across China. Simulations of the dynamic downscaling by the regional climate model
RegCM4 under the RCP4.5 (Representative Concentration Pathway) scenarios have
shown that the air environment carrying capacity tends to decrease, and the weak
ventilation days tend to increase, in the 21$^{st}$ century across China, suggesting an
increase in the haze pollution potential compared to the current state (Han et al., 2017).
Furthermore, Cai *et al.* (2017) projected that the days conducive to severe haze
pollution in Beijing would increase by 50% at the end of the 21$^{st}$ century (2050-2099)
under the RCP8.5 scenarios compared to the historical period.

These qualitative estimations of the haze pollution response to climate changes

generally derived from the *potential* changes of the corresponding meteorological
conditions indirectly. No studies to date quantitatively assessed the simulated PM
directly. How the fine particulate matter pollution changes in response to the global
warming in China has not been thoroughly elucidated to date. This study particularly
focuses on the anthropogenic $PM_{2.5}$ loading and its response to the future warming. In
this study, the large ensemble simulations from the Community Earth System Model
Version 1 (CESM1) throughout the 21[st] century that are induced by increasing
greenhouse gases (GHG) emissions along the trajectory RCP8.5 but retaining the
emissions of aerosols and/or their precursors fixed at the year of 2005 level
(RCP8.5_FixAerosol2005; Xu and Lamarque, 2018) will be utilized.

## 2. Data and methods

### 2.1 $PM_{2.5}$ observational datasets

Surface hourly $PM_{2.5}$ concentration data released since 2013 are taken from the
website of the Ministry of Environmental Protection (http://106.37.208.233:20035),
which covers 1602 sites across China. The duration of available datasets varies across
sites because of the gradual development of the monitoring network in recent years. In
our study region of eastern China (east to 100 $^o$E), there are 1263 sites remaining after
the sites with missing values were removed during 2015-2017. Additionally, surface
daily $PM_{2.5}$ concentrations for the Beijing, Shanghai, Guangzhou, and Chengdu cities
that had relatively longer monitoring times are also collected from the U.S. Beijing
Embassy (http://www.stateair.net/web/historical/1/1.html).

### 2.2 CESM1 model simulations

The CESM1 is an Earth system model involving the atmosphere, land, ocean,
and sea-ice components with a nominal $1°$ by $1°$ horizontal resolution (Hurrell et al.,
2013). The RCP8.5_FixAerosol2005 simulations are forced by the RCP8.5 scenario,
but all emissions of sulfate ($SO_4$), black carbon (BC) and primary organic matter
(POM), and secondary organic aerosols (SOA; or their precursors) and atmospheric
oxidants are fixed at the present-day level (2005). These simulations include 16
ensemble members, differing solely in their atmospheric initial conditions with a tiny
random temperature difference (order of $10^{-14}$ ℃; Kay et al., 2015). For comparison,
the CESM1 large ensemble consists of 35-member simulations that forced by the
RCP8.5 scenario are also employed here. Using these relatively large ensembles can
substantially reduce the contribution of natural variability of the climate system to the
result estimation (Xu and Lamarque, 2018).
For the aerosol emission in the RCP scenarios database, just its decadal change is
considered rather than the emission at a single year (Lamarque et al., 2011). Here, the
years of 2006-2015 are considered as the reference period in the
RCP8.5_FixAerosol2005 simulations. The differences of the mean climates from the
reference period are largely due to the increase in GHG emissions and are not
attributed to the decline in aerosol emissions, as specified in RCP8.5. The changes of
anthropogenic $PM_{2.5}$ loadings and anthropogenic air pollution days in our study are
thus only a result of the GHG-induced climate change, rather than changes in aerosol
emission. Note that just four species of $PM_{2.5}$ components that show a substantial
threat to public health are considered here for analysis, including SO4, BC, POM, and
SOA from the CESM1 simulations.

## 2.3 Definition of the fraction of attributable risk


The influences of the GHG-induced climate changes on the anthropogenic air
pollutions in China are investigated using the metric of the fraction of attributable risk
(FAR), which has been widely used for attribute analyses of climate extreme changes
(Chen and Sun, 2017; Stott et al., 2004). FAR is defined as the $1-P_0/P_1$, where $P_0$ is
the probability of exceeding a certain threshold during the reference period and $P_1$ is
the probability exceeding the same threshold during a given period. FAR thus presents
the quantitative estimations of effects of the GHG-induced climate changes on the
anthropogenic air pollutions.

## 2.4 Definition of stagnation days


The changes of the stagnation days that were induced by the increase of GHG
emissions are also evaluated in our study to explore the possible impact of climate
change on the anthropogenic air pollutions. The day is considered to be stagnant when
the daily mean near-surface wind speed is less than 3.2 m/s, the daily mean 500-hPa
wind speed is less than 13 m/s, and the daily accumulated precipitation is less than 1
mm (Horton et al., 2012). Early studies have suggested that this air stagnation
definition might not be applicable for China to represent the air pollution condition
under the seasonal scales (Feng et al., 2018; Wang et al., 2018). However, the annual
mean stagnation generally presents good agreement with that of air pollution across
China (Huang et al., 2017; 2018). The changes in the annual mean states of air
stagnations over China at the end of $21^{st}$ century will thus be discussed in the
following.

## 3. Results

**3.1 Observational changes in PM$_{2.5}$ pollutions**

The days of severe haze pollution increased over the past several decades across eastern China, particularly for the episodes of January 2013, December 2015, and December 2016, when several severe haze alerts were reached. High PM$_{2.5}$ loading was centralized over the Jing-Jin-Ji (JJJ) region, Shangdong, and Henan provinces, as well as the Sichuan Basin (SCB, Fig. 1a). The annual mean PM$_{2.5}$ mass concentrations for most sites over these regions exceed 75 μg/m$^3$. According to the statistics, there are approximately 95% sites where the annual mean PM$_{2.5}$ concentration exceeded the WHO recommended 24-hour standard (25 μg/m$^3$) across eastern China, and there are 65 sites centralized by Beijing, where the annual mean PM$_{2.5}$ concentration was larger than 75 μg/m$^3$, which would present the possibility of exposing people to serious health hazards (World Health Organization, 2014).

Regarding the four economic zones of Beijing, Shanghai, Guangzhou, and Chengdu cities over China, serious PM$_{2.5}$ pollution can be expected in recent years, especially for the Beijing and Chengdu regions (Fig. 1). Taking Beijing as an example, the annual mean PM$_{2.5}$ concentration was stably exceeding 100 μg/m$^3$, and more than a half of the year had experienced severe air pollution (> 75 μg/m$^3$) before 2013. Since 2013, China's State Council released its Air Pollution Prevention and Control Action Plan, which requires the key regions, including the JJJ, the Yangtze River Delta (YRD), and the Pearl River Delta (PRD) to reduce their atmospheric levels of PM$_{2.5}$ by 25%, 20%, and 15%, respectively, by the end of year 2017 (State Council,

2013). Effort is obvious, and the PM$_{2.5}$ loading and the air pollution days present
sharp decreases in recent years. However, the strict emission policies substantially
cost the economic development, which cannot meet the current requirement of the
rapid development of China. Thus, scientifically quantifying the roles of
anthropogenic emissions and climate changes shows great importance for seeking the
balance between socioeconomic development and emission reduction.
**3.2 Simulated changes in anthropogenic PM$_{2.5}$ pollutions**
A strong spatial correlation (0.69) is found for the annual mean PM$_{2.5}$
concentration between the site observation and median ensemble of CESM1
simulations over eastern China (Fig. S1). The high concentrations across eastern
China, including the regions centralized by Beijing and Chengdu, are reasonably
reproduced. However, a negative bias is obvious. Early studies (Li et al., 2016; Yang
et al., 2017b; c) have documented that this low bias of aerosol concentration simulated
by models is much more complicated in China and the causes mainly involve the
uncertainties from aerosol emission amount, emission injection height, lack of nitrate,
aerosol treatment in model as well as the coarse model resolution.
The median ensemble-mean change of the PM$_{2.5}$ surface concentration presents
strong regional dependence across China with significantly decreasing trends over the
southeastern part of eastern China and significantly increasing trends over the other
regions throughout the 21$^{st}$ century (Fig. S2), even though the emissions are constant
throughout the experiment. The regional differences in the total PM$_{2.5}$ changes are
mainly due to SO$_4$, which can account for approximately 50% of the total PM$_{2.5}$ mass
(Xu and Lamarque, 2018). The species of BC and POM are reported to significantly
increase in the 21$^{st}$ century across eastern China, although the aerosol emissions were
fixed at the level in 2005. Figure 2 presents the simulated PM$_{2.5}$ loadings from the
CESM1 model, in terms of column burden and surface concentration, are significantly
increasing throughout the 21$^{st}$ century. The increase in the total PM$_{2.5}$ is
approximately 8% for the column burden and 2% for the surface concentration at the
end of the 21$^{st}$ century (2090-2099) with respect to the current state (2006-2015).
These increasing trends of PM$_{2.5}$ loadings are mainly due to the significant increases
of the major PM$_{2.5}$ species, except for SOA, in which the surface concentration
presents a slight decrease. Furthermore, the increases of all major PM$_{2.5}$ species in
terms of column burden (BC: 11%, SO$_4$: 6%, SOA: 11%, and POM: 11%) show
stronger than the surface concentration (BC: 4%, SO$_4$: 2%, SOA: -1%, and POM:

4%).

For comparison, we also evaluated the future changes of PM$_{2.5}$ concentrations

and the associated species along the RCP8.5 forcing trajectory from the large
ensemble simulations of CESM1 (Figure not shown). Different from changes of
aerosol concentrations under the fixed aerosol simulations, the PM$_{2.5}$ concentrations
and the associated species present uniformly decreasing trends across eastern China
from the simulations along the RCP8.5 forcing. The decreasing trends in the RCP8.5
simulations are mainly attributed to the prescribed decrease of aerosol forcing in the
future in RCP database (Xu and Lin, 2017). The climate change induced by the
GHG-warming might exacerbate the air pollution, but the impacts cannot compensate
the prescribed decreasing trend of aerosol concentration.
As mentioned above, the PM$_{2.5}$ surface concentration in the two economic zones
of YRD and PRD present a negative response to the GHG-induced warming, while
the corresponding column burden shows significantly increasing trends (Fig. S3). The
decreases of the surface concentration over these two zones are primarily contributed
by the changes of SO$_4$ and SOA, while there are no obvious trends for BC and POM
(Figs. S4-S7). The robust response of the increased surface wind speed and decreased
upper-level wind speed to GHG warming can be partly responsible for the changes of
the major PM$_{2.5}$ species in these two zones, which will be further discussed. Over the
zones of JJJ and SCB, both the PM$_{2.5}$ concentrations and the associated major PM$_{2.5}$
species present the significantly rising trends throughout the 21$^{st}$ century. For the
surface concentration, PM$_{2.5}$ is reported to increase by 3% and 4% in the regions of
JJJ and SCB, respectively, at the end of the 21$^{st}$ century. The BC is reported to
increase by 4% and 8% for JJJ and SCB, respectively. The other species, such as SO$_4$
and POM, increase by 4% and 4%, respectively, in the JJJ regions and by 2% and 9%,
respectively, in SCB regions. Relatively stronger responses can be seen in changes of
the column burden for all major species (Figs. S4-S7). The increased concentrations
of PM$_{2.5}$ species finally result in significantly increasing trends of the total PM$_{2.5}$
loading over these two regions, which will present a more direct effect on human
health.
The increase in PM$_{2.5}$ surface concentration throughout the 21$^{st}$ century
substantially leads to the significant increase of the light anthropogenic PM$_{2.5}$
pollution days ($PM_{2.5} > 25$ μg/m$^3$) across the northwestern part of eastern China (Fig.
3). Due to the decrease of $PM_{2.5}$ concentration over the southeastern part of eastern
China, the light anthropogenic air pollution days can be expected to decrease in this
region. Estimation shows that the number of the light air pollution days would be
decreased by approximately 10 days at the end of the 21$^{st}$ century with respect to the
early period of this century in the region. However, the annual mean light air pollution
days is reported to increase averaged over the eastern China at the end of this century
despite the aerosol emission is constant throughout the experiment. In contrast to the
light air pollution days, the severe anthropogenic air pollution days ($PM_{2.5} > 75$ μg/m$^3$)
show a positive response to the GHG-induced warming across eastern China,
particularly for the regions around JJJ in which the high $PM_{2.5}$ concentration was
localized (Fig. 3). The severe air pollution days is estimated to increase by more than
2 days at the end of this century when compared to the early period over this region.
Considering the underestimation in aerosol concentration by CESM1 model in China,
the percentile threshold metric is also applied here to estimate the future changes in
light (90th) and severe (99th) air pollution days. Similar results can be obtained (Fig.
S8).
**3.3 Attributable changes due to GHG warming**
Although the aerosol emission was constant throughout the experiment, our
study reveals that the $PM_{2.5}$ loadings and their associated pollution days still present
increases throughout the 21$^{st}$ century, primarily resulting from the impact of climate
change induced by GHG warming. One may ask how large a contribution the climate
change exerts on the changes in anthropogenic air pollution. To quantitatively address
this issue, the framework of the "Fraction of Attributable Risk (FAR)" that has been
widely used for attribute analyses of climate extreme changes (Chen and Sun, 2017;
Stott et al., 2004) is employed in this study.
Figure 4 shows the percentage changes of the anthropogenic air pollution days
throughout the 21$^{st}$ century over eastern China and their associated FAR variations.
The regional averaged anthropogenic air pollution days present an obvious increase in
the 21$^{st}$ century as addressed above. Correspondingly, synchronous increasing trends
can be found in FAR for both light and severe anthropogenic air pollution days. For
the light pollution days, FAR is estimated to be 28% at the end of the 21$^{st}$ century,
implying that approximately 28% of the pollution days are contributed by the climate
change that was induced by GHG warming. For the severe pollution days, FAR shows
a relatively smaller value of approximately 11%. Furthermore, the high FAR values
are mainly located over the regions of high $PM_{2.5}$ loadings concentrated over eastern
China, suggesting considerably stronger effects of climate changes in these regions.
Note that the FAR values estimated in this research may be underestimated because
the GHG-induced warming impact was involved in the selected reference period that
resulted in the overestimation of the probability of anthropogenic air pollution days.
**3.4 Effects of the changes in meteorological conditions**
We further examined the changes of meteorological conditions induced by the
GHG warming that alternatively exerted effects on air pollution. Our results show that
the local boundary layer height presents as higher under the warming scenario (Fig.
5a), which benefits the vertical transport of the air pollutant.

However, a robust negative response of the horizontal advection to the

GHG-induced warming across eastern China can be found in the troposphere (Fig. 5b,
c), facilitating air pollutant accumulation. The change of surface wind speed in
response to the GHG warming is highly similar with the variation of $PM_{2.5}$ surface
concentration, with wind speed increasing in the southeastern part of eastern China
and decreasing in the northwestern part. Variations of surface wind speeds are thus
mainly responsible for the changes of $PM_{2.5}$ surface concentration over eastern China.
Different responses can be found for the tropospheric upper-level wind speeds, which
are reported to substantially decrease. These decreases would directly result in
significant increases of the stagnation days over eastern China, particularly over the
northern region and SCB (Fig. 6). The decreasing trend of wind speed in the 21[st]
century across China not only exists in CEMS1 model, but also happens in the other
global climate models that participated in Coupled Model Intercomparison Project
Phase 3 (CMIP3) and CMIP5 (Jiang et al., 2010a; Mclnnes et al., 2011), as well as in
regional climate models (Jiang et al., 2010b).

In response to the GHG-induced warming, the stagnation days over eastern

China are estimated to increase by 6% at the end of 21[st] century with respect to the
current period. For the specific economic zones, the stagnation days over the SCB and
JJJ regions show considerably stronger rising trends, while relatively weaker increases
are observed over the YRD and PRD regions. The number of stagnation days is
estimated to increase by 13% and 6% at the end of the 21[st] century for the SCB and
JJJ regions, respectively. Briefly, though the atmospheric stratification appears to be
considerably more unstable in response to the GHG warming, the weakened
horizontal advection would substantially increase the stagnation days over eastern
China, which provides a beneficial background for the air pollutant accumulation and
further increases the occurrence probability of the anthropogenic air pollution events.

Early studies have documented a significant increase in total precipitation across

China due to the GHG-induced warming (Chen, 2013; Li et al., 2018; Wang et al.,
2012), which seems to represent a conflict with the increase of the anthropogenic air
pollution days. To resolve this issue, the precipitation changes in terms of light
precipitation days (daily accumulated precipitation < 10 mm) and heavy precipitation
days (> 10 mm) are further examined (Fig. 5d, e). Clearly, the heavy precipitation
days present an increase, while the light precipitation days show a decrease, across
eastern China in response to the warming. Though the precipitation shifts toward
heavy precipitation events, its cleansing impact on air pollutants has not increased
because an increase in heavy precipitation days appears to be insufficient to further
enhance the wet removal ability (Xu and Lamarque, 2018). In contrast, the decrease in
light precipitation days substantially weakens the wet deposition of air pollutants,
leading to the increase of the $PM_{2.5}$ loading, as well as anthropogenic air pollution
days. The future changes of precipitation days present much robust. Both the
increasing trends of heavy precipitation days and the decreasing trends of light
precipitation days are also obvious across China simulated by the CMIP5 models
(Chen and Sun, 2013; 2018), as well as the regional climate models (Gao et al., 2012).

## 4. Conclusions

The world is predicted to experience increased disasters, such as heat waves, flash floods, and storms, due to the continuous global warming induced by the GHG increase. The research question we aim to address in this study is how the GHG warming would affect the anthropogenic $PM_{2.5}$ pollutions across China. Our evaluations show that the anthropogenic $PM_{2.5}$ loadings, as well as the anthropogenic $PM_{2.5}$ pollution days, would increase under the global warming conditions even the aerosol emissions fixed at current levels. More stringent regulations are thus suggested for regional aerosol emissions to maintain the air quality standard as the current state.

The climate changes induced by GHG warming exert their effects on the anthropogenic air pollutions across eastern China via two ways that are of interest in this study. First, the weakened tropospheric wind speed induced by the GHG warming would result in a decrease of the horizontal advection and lead to an increase in the number of stagnation days, facilitating the local accumulation of air pollutants. Second, the number of light precipitation days would decrease due to GHG-induced warming, although the total precipitation would clearly increase across China. This shift toward more no-rainfall days would further weaken the wet deposition of $PM_{2.5}$ pollutants. Thus, the increased stagnation days and decreased light precipitation days provide a beneficial background for the occurrence of anthropogenic air pollution. Of course, under the warming scenarios, a large discrepancy exists among the different meteorological processes that benefit the air pollutions at the current state, leading to

the fuzzy recognition of air pollution change. For example, the boundary layer height
shows an increase in response to the GHG warming that may strengthen the vertical
dissipation of air pollutants. Thus, more studies are suggested in the future to further
understand the mechanisms governing air quality across China.


## Author contributions

H. P. Chen and H. J. Wang designed the research; H. P. Chen analyzed the data.
All the authors discussed the results and wrote the paper.

## Competing interests

The authors declare that they have no conflict of interest.

## Acknowledgements

This work is jointly supported by the National Key Research and Development
Program of China (Grant No: 2016YFA0600701), the National Natural Science
Foundation of China (Grant No: 41421004), and the CAS-PKU Joint Research
Program.

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

**Figure captions**

**Figure 1. Observed PM$_{2.5}$ pollution conditions over eastern China during the past years.** (a) Annual averaged PM$_{2.5}$ concentration ($\mu$g/m$^3$) for the years of 2015-2017. (b) Variations of annual averaged PM$_{2.5}$ concentration (green bars) in Beijing city and the corresponding number of the severe PM$_{2.5}$ pollution days (red bars). The severe pollution days are defined as the daily averaged PM$_{2.5}$ concentration exceeding 75 $\mu$g/m$^3$. (c), (d), and (e) are similar to (b), but for the results of Shanghai, Guangzhou, and Chengdu city, respectively.

**Figure 2. Plots of future changes of the total PM$_{2.5}$ as well as its associated species averaged over eastern China** in terms of the surface concentration ($\mu$g/m$^3$, right axis in red) and column burden (mg/m$^2$, left axis in blue) from the simulations of the RCP8.5_FixAerosol2005 experiment. (a) PM$_{2.5}$, (b) BC, (c) SO$_4$, (d) POM, and (e) SOA. Ensemble variance (1 sigma) for surface concentration is shown in red shadings.

**Figure 3. Changes of the anthropogenic PM$_{2.5}$ pollution days across eastern China from the RCP8.5_FixAerosol2005 experiment**. The top panel (a, b) shows the changes of light air pollution days ($> 25$ $\mu$g/m$^3$) and the bottom panel (c, d) shows the results of severe air pollution days ($> 75$ $\mu$g/m$^3$). The left panel (a, c) illustrates the annual averaged severe pollution days in 2006-2015 and the right panel (b, d) shows changes of the pollution days at the end of the 21$^{st}$ century with respect to 2006-2015. Dots in (b) and (d) mean the changes are significant at the 95% confidence level using Student T-test for all years and ensembles. Units: days.

**Figure 4. Attributable changes of anthropogenic air pollution days to the**
**increased greenhouse gases emissions.** (a) Spatial distribution of FAR for the
changes of severe $PM_{2.5}$ pollutions ($> 75$ μg/m$^3$) at the end of the 21$^{st}$ century over
eastern China. (b) Regional averaged relative changes of air pollution days (left axis
in red; $> 25$ μg/m$^3$) and the corresponding variation of FAR (right axis in blue).
Ensemble variance (1 sigma) for the relative changes of pollution days is shown in red
shadings. (c) is similar to (b), but for the severe $PM_{2.5}$ pollution days. Units: %.
**Figure 5. Simulated changes in weather conditions of the air pollutions across**
**eastern China due to the GHG-induced warming.** (a) Changes of the planetary
boundary layer height (PBLH) at the end of the 21$^{st}$ century with respect to the years
of 2006-2015 from the RCP8.5_FixAerosol2005 experiment. (b) and (c) are similar to
(a) but for the wind speed at near-surface and 500-hPa levels, respectively. (d)
Changes in the light precipitation days (daily accumulated precipitation $< 10$ mm) at
the end of the 21$^{st}$ century with respect to the current state. (e) is similar to (d) but for
the heavy precipitation days ($> 10$ mm). Dots in the figure mean the changes are
significant at the 95% confidence level using Student T-test for all years and
ensembles. Units: %.
**Figure 6. Changes in the stagnant conditions across China due to the**
**GHG-induced warming.** (a) Distribution of the relative changes of the stagnation
days at the end of the 21$^{st}$ century against the current state (2006-2015). Dots mean
the changes are significant at the 95% confidence level using Student T-test for all
years and ensembles. (b) Variations of the regional averaged stagnation days over
eastern China. Ensemble variance (1 sigma) is shown in red shadings. (c), (d), (e), and
(f) are similar to (b), but for the results of four Chinese economic zones, i.e., JJJ, YRD,
PRD, and SCB. Units: %.


**Figures**

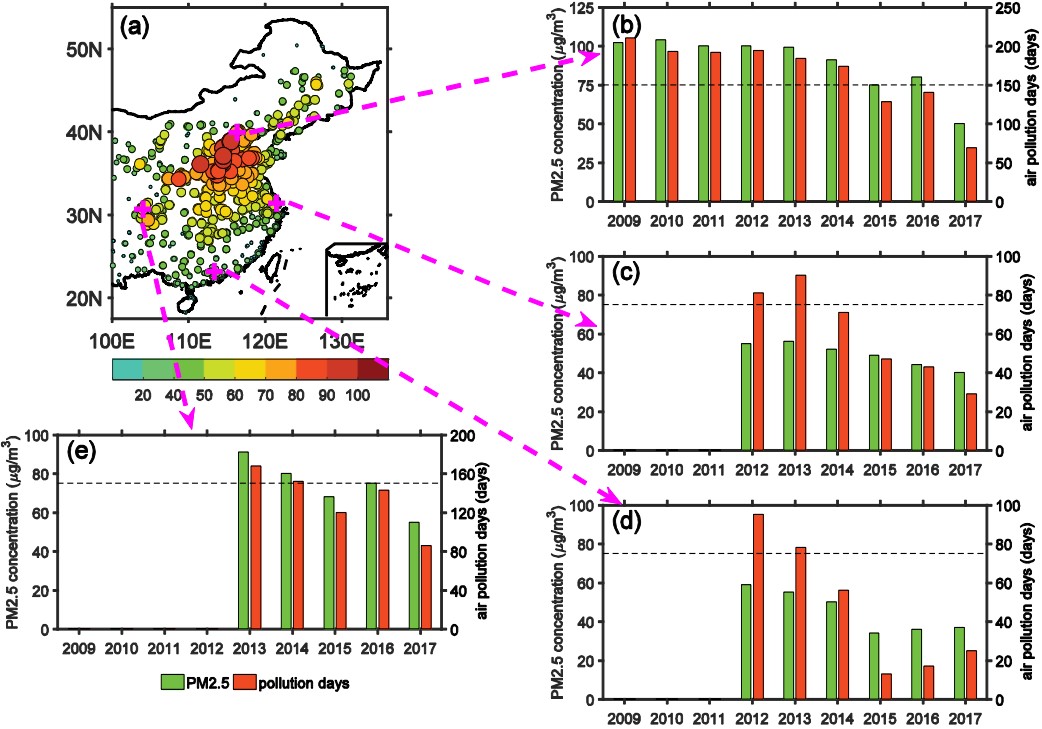


**Figure 1. Observed PM$_{2.5}$ pollution conditions over eastern China during the past**

**years.** (a) Annual averaged PM$_{2.5}$ concentration (μg/m$^3$) for the years of 2015-2017.

(b) Variations of annual averaged PM$_{2.5}$ concentration (green bars) in Beijing city and

the corresponding number of the severe PM$_{2.5}$ pollution days (red bars). The severe

pollution days are defined as the daily averaged PM$_{2.5}$ concentration exceeding 75

μg/m$^3$. (c), (d), and (e) are similar to (b), but for the results of Shanghai, Guangzhou,

and Chengdu city, respectively.

590

591

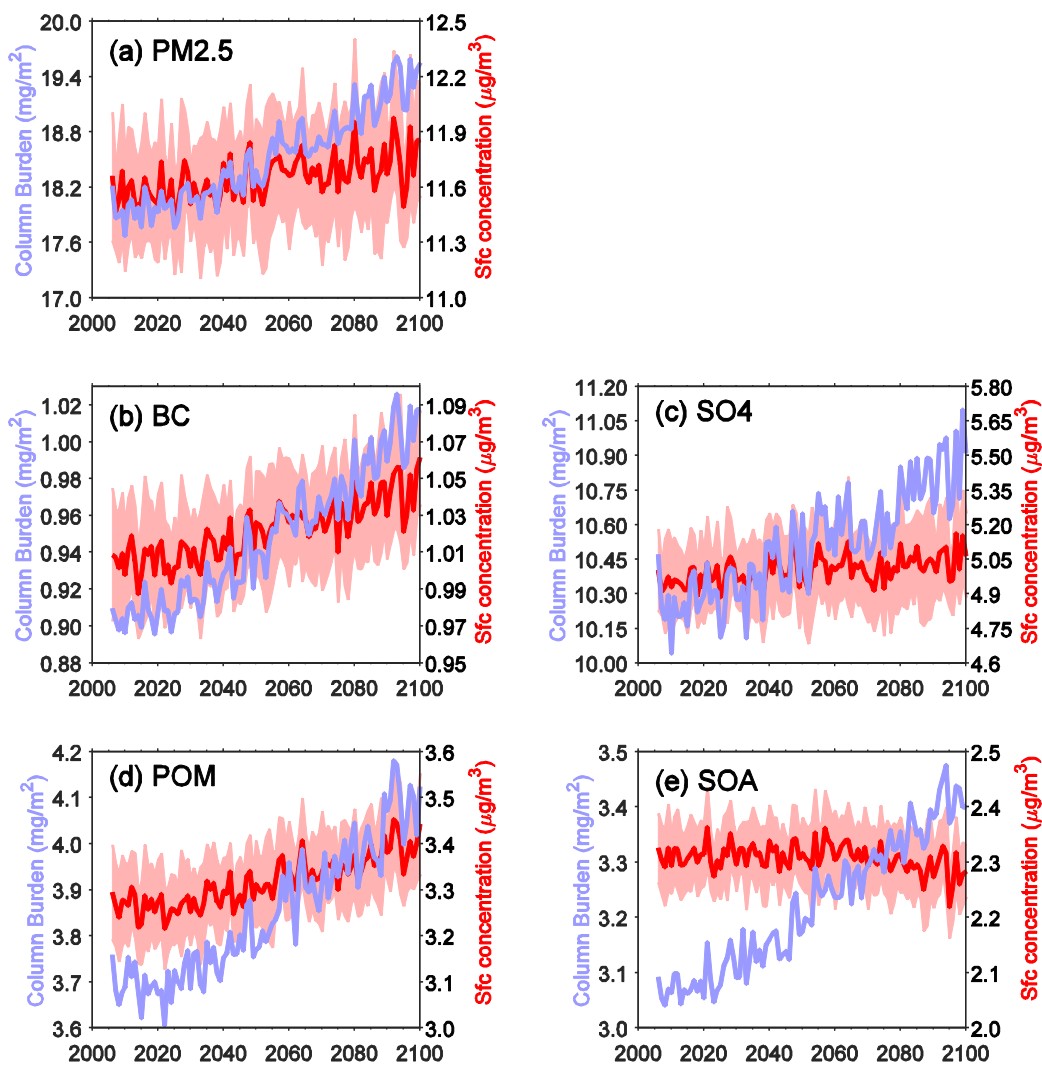

**Figure 2. Plots of future changes of the total PM$_{2.5}$ as well as its associated species averaged over eastern China** in terms of the surface concentration ($\mu$g/m$^3$, right axis in red) and column burden (mg/m$^2$, left axis in blue) from the simulations of the RCP8.5_FixAerosol2005 experiment. (a) PM$_{2.5}$, (b) BC, (c) SO$_4$, (d) POM, and (e) SOA. Ensemble variance (1 sigma) for surface concentration is shown in red shadings.

598

599

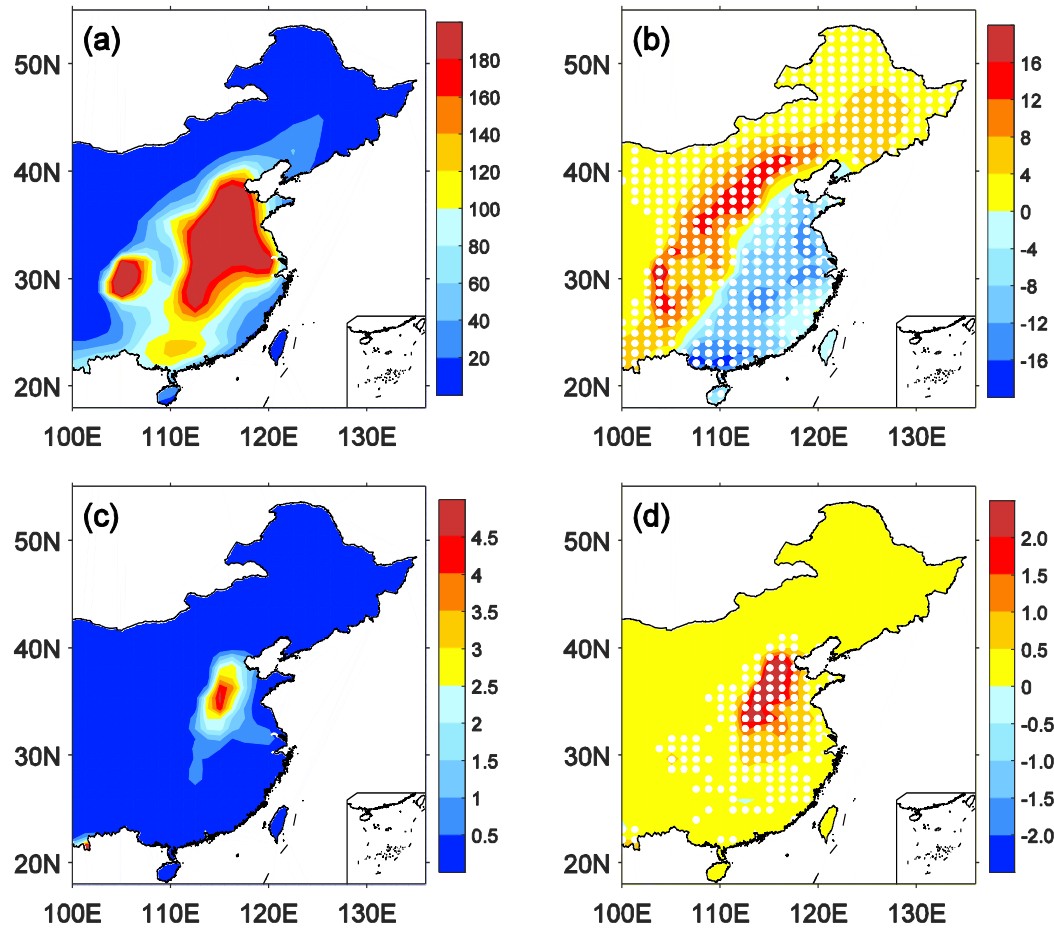

**Figure 3. Changes of the anthropogenic PM$_{2.5}$ pollution days across eastern China from the RCP8.5_FixAerosol2005 experiment**. The top panel (a, b) shows the changes of light air pollution days (> 25 μg/m$^3$) and the bottom panel (c, d) shows the results of severe air pollution days (> 75 μg/m$^3$). The left panel (a, c) illustrates the annual averaged severe pollution days in 2006-2015 and the right panel (b, d) shows changes of the pollution days at the end of the 21$^{st}$ century with respect to 2006-2015. Dots in (b) and (d) mean the changes are significant at the 95% confidence level using Student T-test for all years and ensembles. Units: days.

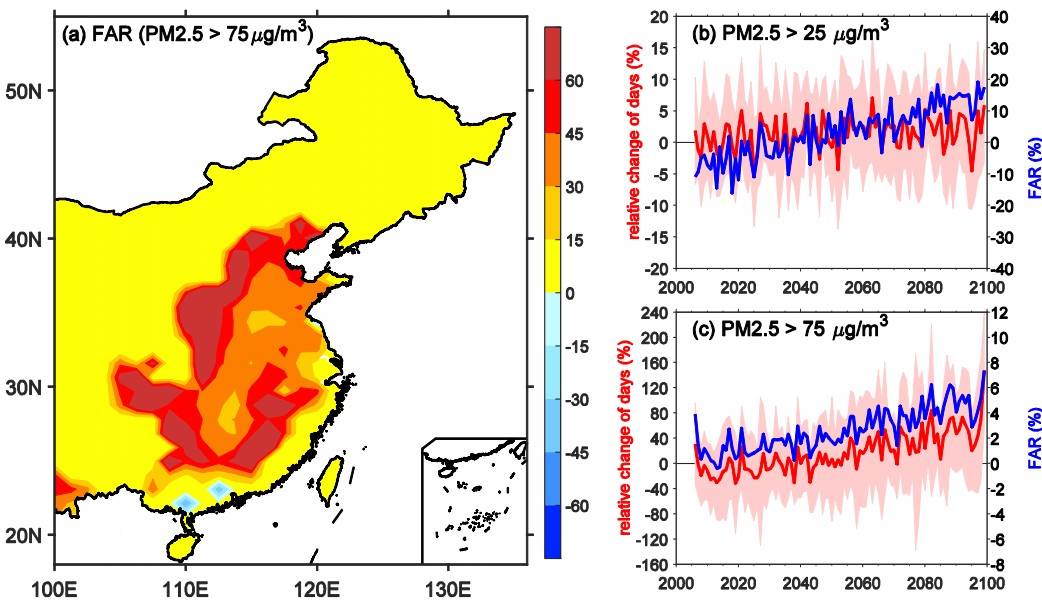

611

**Figure 4. Attributable changes of anthropogenic air pollution days to the increased greenhouse gases emissions.** (a) Spatial distribution of FAR for the changes of severe $PM_{2.5}$ pollutions ($> 75$ μg/m$^3$) at the end of the 21$^{st}$ century over eastern China. (b) Regional averaged relative changes of air pollution days (left axis in red; $> 25$ μg/m$^3$) and the corresponding variation of FAR (right axis in blue). Ensemble variance (1 sigma) for the relative changes of pollution days is shown in red shadings. (c) is similar to (b), but for the severe $PM_{2.5}$ pollution days. Units: %.

619

620

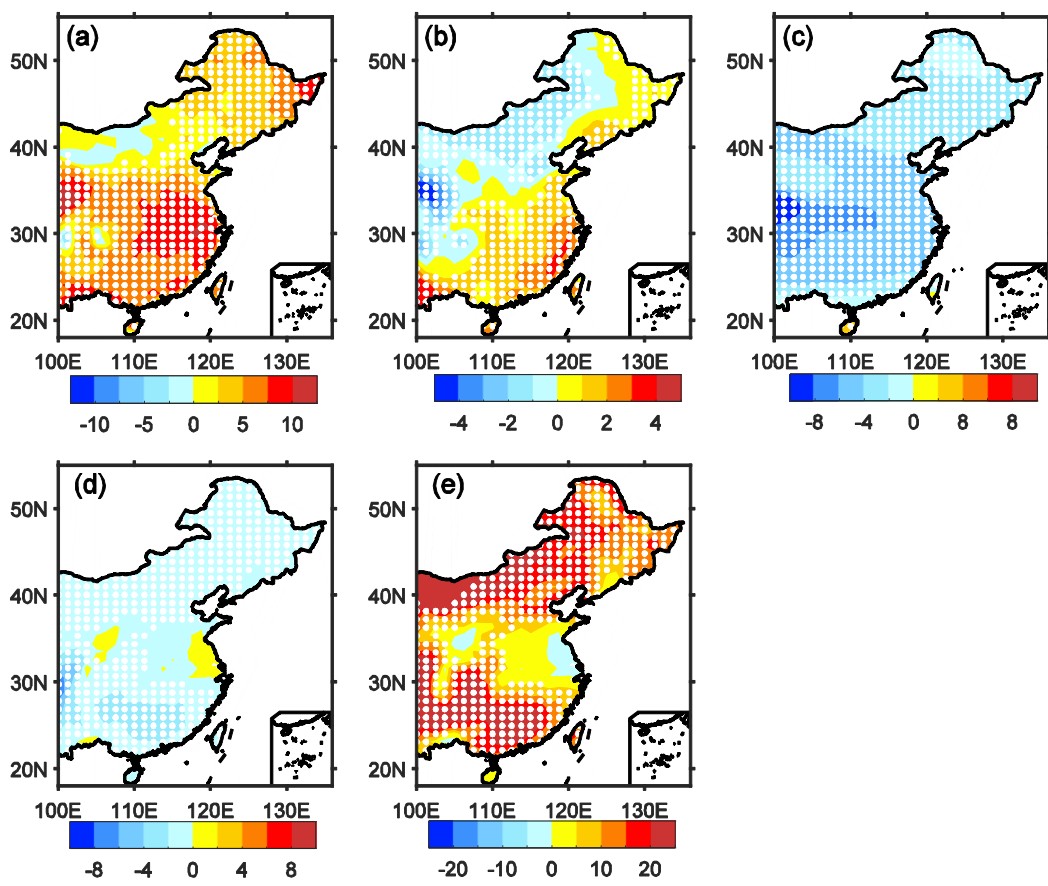

**Figure 5. Simulated changes in weather conditions of the air pollutions across eastern China due to the GHG-induced warming.** (a) Changes of the planetary boundary layer height (PBLH) at the end of the 21[st] century with respect to the years of 2006-2015 from the RCP8.5_FixAerosol2005 experiment. (b) and (c) are similar to (a) but for the wind speed at near-surface and 500-hPa levels, respectively. (d) Changes in the light precipitation days (daily accumulated precipitation < 10 mm) at the end of the 21[st] century with respect to the current state. (e) is similar to (d) but for the heavy precipitation days (> 10 mm). Dots in the figure mean the changes are significant at the 95% confidence level using Student T-test for all years and ensembles. Units: %.

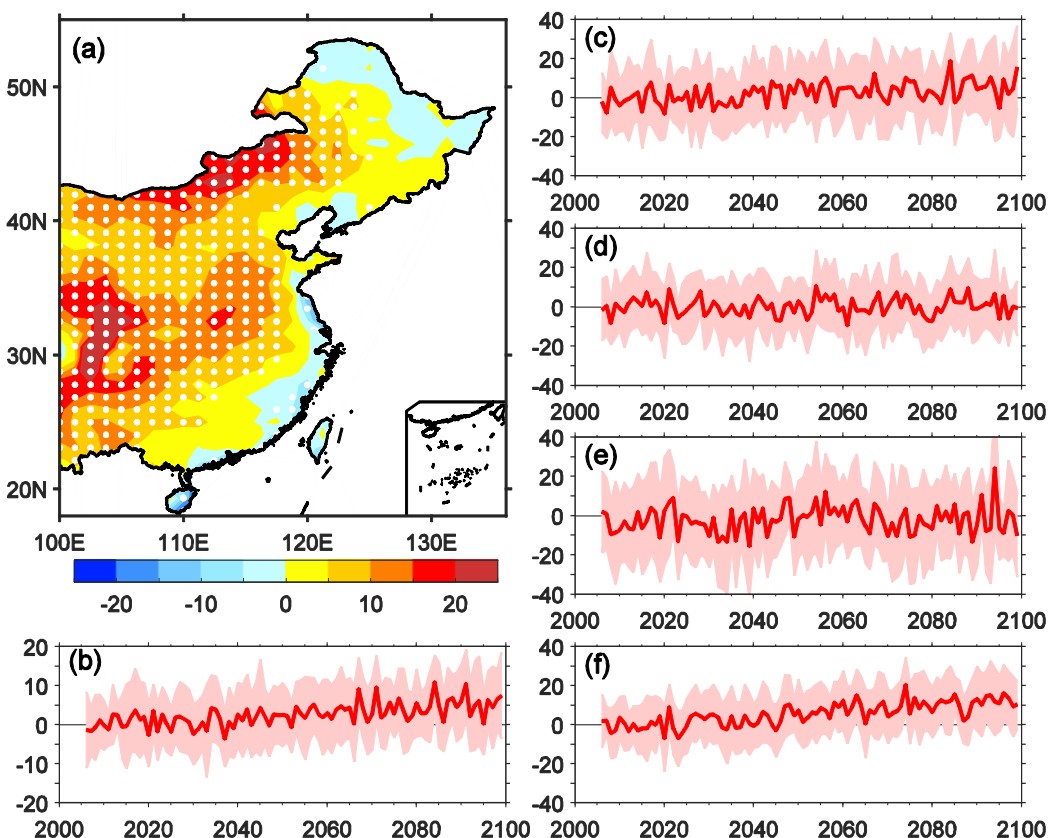

633

**Figure 6. Changes in the stagnant conditions across China due to the GHG-induced warming.** (a) Distribution of the relative changes of the stagnation days at the end of the 21$^{st}$ century against the current state (2006-2015). Dots mean the changes are significant at the 95% confidence level using Student T-test for all years and ensembles. (b) Variations of the regional averaged stagnation days over eastern China. Ensemble variance (1 sigma) is shown in red shadings. (c), (d), (e), and (f) are similar to (b), but for the results of four Chinese economic zones, i.e., JJJ, YRD, PRD, and SCB. Units: %.

642

643