# Peer review of "Anthropogenic Fine Particulate Matter Pollution Will Be Exacerbated in Eastern"

_Atmospheric Chemistry and Physics, 2018_

## Referee Comment (RC1) · Anonymous Referee #2 · 17 Aug 2018

This study examined changes in PM2.5 and pollution days in China under greenhouse gas warming conditions using century-long CESM large ensemble simulations. They found that increases in PM2.5 concentration and pollution days during 2005–2100 with fixed anthropogenic aerosol emissions and changes in winds and light precipitation could explain it. The topic is interesting and suit for ACP journal. However, the results are not convincible, and manuscript needs major revision before it can be considered to be accepted in ACP.

My main concern is that the authors found decreases in PM2.5/SO4/SOA concentration (Figure S2) and moderate pollution days (Figure S8) over YRD and PRD, but they claimed an increase in pollution as a whole under GHG warming in the manuscript. It looks more like a decrease in pollution over eastern China to me. It is OK to present different trend pattern for different regions. Although the severe pollution days show increase over eastern China, the mean severe pollution days is less than 5/365 and the change is around 2/365 days. These values are too small. I would guess it is more like a model noise. The model strongly underestimates aerosol concentration in China. The author may consider using to a lower threshold. The author stated 'with an increase of approximately 68%'. It is unfair since the PM2.5 concentration only increases by 2%. The mechanism of the aerosol change is also unclear. Why SO4/SOA decreases while BC/POM increases in southeastern China?

Another main concern is that the changes in aerosol concentrations are too small between 2090–2099 and 2006–2015. The results are not convincing without significant test. The authors should add test (e.g., t-test for years and ensembles) to the results and figures to prove that the small values are not accidental. Changes in meteorological parameters also needs significant test. In addition, CESM model strongly underestimates aerosol concentration over China. Using absolute concentrations may cause problem, for example PM2.5>25/75. The authors should be caution about it.

Minor comments:

Line 27: The authors only used one model configuration with fixed anthropogenic emissions. It can definitely be used to rule out the influence of changing emissions. However, for a supplement of the story, I suggest comparing the role of GHG warming and aerosol emission change. I know it is hard to do another simulation, but the authors can easily scale aerosol concentration by emissions using RCP8.5 scenario and roughly predict aerosol concentration under future emissions.

Line 83: The highlight of this study, I guess, it the PM2.5 concentration under global warming. But the PM2.5 change is too small, and the authors showed pollution days instead (68% in abstract), which may not be appropriate and cannot be distinguished to precious studies. The authors may consider re-organize the findings and change the way of presentation.

Line 125-133: I don't understand what is fraction of attributable risk. The author can give an example and illustrate what is it used for.

Line 134: The author used stagnation day defined by Horton et al. (2012). It is definition suit to stagnation in China? Is there any previous study used it for stagnation in China? If not, the authors have to evaluate it using historical data of China.

Line 170: Spatial correlation over eastern China?

Line 174: I don't agree the bias is primarily due to missing species. The bias of aerosol concentration is more complicated in China, which has been reported in many previous studies (Yang et al., 2017a,b). The causes include uncertainties in aerosol emission amount, emission injection height, course model resolution, lack of nitrate, aerosol treatment in model (e.g., aging processes, chemistry, wet removal)…

Line 187: 2% is too small. The authors can focus on different regions and species.

Line 190: Add SOA in Figure 2.

Line 200: Do the future changes in meteorology (winds, precipitation) over China also exist in other models? At least add literatures.

Line 217: I don't think it is a good idea to emphasize and severe days and use 'robust response'. First, PM2.5 > 75 is suit for observations. The simulated PM2.5 is only 1/3 of the observation value. Second, I don't think the change will large than the standard deviation of severe days for different ensemble simulations and years.  BTW, do you mean 'positive' response.

Line 226: 'PM2.5 loadings and their associated pollution days still present significant increases'. As mentioned above, I don't think this statement is correct.

Line 228-246: I don't understand what this section is used for. '28% of the pollution days are contributed by the climate change that was induced by GHG warming.' The authors fixed aerosol emission and all changes (100%) in the model should be due to GHG warming. Even changes in pollution days due to changes in meteorological conditions result from GHG warming.

Line 293-295: Again, 'substantially increase' is not correct.

Reference:

Yang, Y., Wang, H., Smith, S. J., Ma, P.-L., and Rasch, P. J.: Source attribution of black carbon and its direct radiative forcing in China, Atmos. Chem. Phys., 17, 4319-4336, https://doi.org/10.5194/acp-17-4319-2017, 2017.

Yang, Y., Wang, H., Smith, S. J., Easter, R., Ma, P.-L., Qian, Y., Yu, H., Li, C., and Rasch, P. J.: Global source attribution of sulfate concentration and direct and indirect radiative forcing, Atmos. Chem. Phys., 17, 8903-8922, https://doi.org/10.5194/acp-17-8903-2017, 2017.

---

## Referee Comment (RC2) · Anonymous Referee #1 · 20 Aug 2018

Chen et al. attempted to elucidate how PM pollution in eastern China will response to future GHG warming, using a large ensemble of CESM simulations. The authors reported that GHG-induced climate change will increase PM pollution days, especially the most severe polluted days (PM2.5>75 $\mu$g m-3), at the end of 21th century and they argued that reduced tropospheric winds and light precipitation days can be the reasons. Their results are interesting and could deepen our understanding of the impacts of climate change on air quality. The topic is suitable for ACP readers, and this paper is well structured. However, I have some concerns about the linkage between pollution increase and changes in meteorology. The authors need to address the following comments before it can be published.

General Comments:

- The authors found an increase of 68% in the most severe pollution days, with only an increase of 3% for light pollution days, but they attributed such increase to the mean change of future GHG-induced climate change. In statistics, I think the increase in most severe pollution days represents the extreme cases, whose linkage to mean climate change needs to be further explored, or at least discussed.

- The ACCMIP (Lamarque et al., 2013) also archives similar simulations by several climate models. It would be helpful if the authors can compare their results with ACCMIP models. Just a suggestion.

Specific Comments:

-Line 32-34: As indicated above, the authors should take care here.

-Line 134-140: The relationship between air stagnation index used here and PM2.5 pollution in China may be not well correlated (e.g., Feng et al., 2018).

-Line 148 and Figure 1: Why chose a reference concentration of 75 $\mu$g m-3. The annual PM2.5 standard in China is 35 $\mu$g m-3.

-Line 170-172: The correlation is based on what observational and model data. Should make it clear.

-Line 173-175: Same as above, the low bias in model depends on what observational PM2.5 you used. As reported in Li et al. (2016), the RCP emissions for year 2005 underestimated anthropogenic emissions of aerosol precursors over China. Thus, the lower PM2.5 concentration in model could also partly attribute to underestimated emissions.

-Line 182-183: Which region you average the PM2.5 concentration for "eastern China"?

-Line 205: What "SC" shorts for?

Reference:

Lamarque et al., (2013). The Atmospheric Chemistry and Climate Model Intercomparison Project (ACCMIP): overview and description of models, simulations and climate diagnostics. GMD. 6(1), 179-206.

Feng et al., (2018). An air stagnation index to qualify extreme haze events in northern China. JAS.

Li et al., (2016). Implications of RCP emissions on future PM2.5 air quality and direct radiative forcing over China. JGR. 121(21).

---

## Author Comment (AC1) · 12 Sep 2018

**Response to Reviewer #1:**

This study examined changes in PM2.5 and pollution days in China under greenhouse gas warming conditions using century-long CESM large ensemble simulations. They found that increases in PM2.5 concentration and pollution days during 2005–2100 with fixed anthropogenic aerosol emissions and changes in winds and light precipitation could explain it. The topic is interesting and suit for ACP journal. However, the results are not convincible, and manuscript needs major revision before it can be considered to be accepted in ACP.

**Reply:** Greatly thanks for your valuable comments and suggestion, which have been fully considered and corrected in the current manuscript.

My main concern is that the authors found decreases in PM2.5/SO4/SOA concentration (Figure S2) and moderate pollution days (Figure S8) over YRD and PRD, but they claimed an increase in pollution as a whole under GHG warming in the manuscript. It looks more like a decrease in pollution over eastern China to me. It is OK to present different trend pattern for different regions. Although the severe pollution days show increase over eastern China, the mean severe pollution days is less than 5/365 and the change is around 2/365 days. These values are too small. I would guess it is more like a model noise. The model strongly underestimates aerosol concentration in China. The author may consider using to a lower threshold. The author stated 'with an increase of approximately 68%'. It is unfair since the PM2.5 concentration only increases by 2%. The mechanism of the aerosol change is also unclear. Why SO4/SOA decreases while BC/POM increases in southeastern China?

**Reply:**

(1) As you indicated, the changes of $PM_{2.5}$ surface concentration and the corresponding light air pollution days are varying from regions, with a decreasing trend over the southeastern part of eastern China and an increasing trend over the northwestern part. However, the $PM_{2.5}$ loading is estimated to present positive response to the GHG-induced warming that averaged over the region of eastern

China, as well as the light air pollution days. More information can found in the current manuscript.

→[Line 28-34]

Our results show that although the aerosol emission is assumed to be a constant throughout the experiment, anthropogenic air pollution presents positive responses to the GHG-induced warming. The anthropogenic $PM_{2.5}$ concentration is estimated to increase averaged over eastern China at the end of this century, but varying from regions, with an increase over northwestern part of eastern China and a decrease over southeastern part. Similar changes can be observed for the light air pollution days.

→[Line 191-195]

The median ensemble-mean change of the $PM_{2.5}$ surface concentration presents strong regional dependence across China with significantly decreasing trends over the southeastern part of eastern China and significantly increasing trends over the other regions throughout the 21st century (Fig. S2), even though the emissions are constant throughout the experiment.

→[Line 239-248]

The increase in $PM_{2.5}$ surface concentration throughout the 21st century substantially leads to the significant increase of the light anthropogenic $PM_{2.5}$ pollution days ($PM_{2.5} > 25$ $\mu g/m^3$) across the northwestern part of eastern China (Fig. 3). Due to the decrease of $PM_{2.5}$ concentration over the southeastern part of eastern China, the light anthropogenic air pollution days can be expected to decrease in this region. Estimation shows that the number of the light air pollution days would be decreased by approximately 10 days at the end of the 21st century with respect to the early period of this century in the region. However, the annual mean light air pollution days is reported to increase averaged over the eastern China at the end of this century despite the aerosol emission is constant throughout the experiment.

(2) Yes, you are right. The expression referring to the change of severe air pollution days may be inaccurate. We have toned down the related expressions in the current manuscript and more information can be found in the text.

→[Line 34-35]

However, the severe air pollution days is reported to increase across eastern China at the end of this century, particularly around the Jing-Jin-Ji region.

→[Line 248-253]

In contrast to the light air pollution days, the severe anthropogenic air pollution days ($PM_{2.5} > 75$ μg/m$^3$) show a positive response to the GHG-induced warming across eastern China, particularly for the regions around JJJ in which the high $PM_{2.5}$ concentration was localized (Fig. 3). The severe air pollution days is estimated to increase by more than 2 days at the end of this century when compared to the early period over this region.

Another main concern is that the changes in aerosol concentrations are too small between 2090–2099 and 2006–2015. The results are not convincing without significant test. The authors should add test (e.g., t-test for years and ensembles) to the results and figures to prove that the small values are not accidental. Changes in meteorological parameters also needs significant test. In addition, CESM model strongly underestimates aerosol concentration over China. Using absolute concentrations may cause problem, for example PM2.5>25/75. The authors should be caution about it.

**Reply:**

(1) As you suggested, the significant test has been added to the results and figures using Student T-test for years and ensembles. Figures 3, 5, and 6 have been re-plotted. More information can be found in the manuscript.

(2) The changes of air pollution days that defined according to the percentile thresholds (90$^{th}$ and 99$^{th}$) have been also investigated and similar results can be obtained (Figure 1). Thus, the absolute definition is still used in the current manuscript, and some discussions about the percentile definition have been added in the text.

[Figure]

**Figure 1.** Changes of the anthropogenic PM$_{2.5}$ pollution days across eastern China from the RCP8.5_FixAerosol2005 experiment. The pollution days is defined as the daily PM$_{2.5}$ surface concentration exceeding the 90th percentile threshold that estimated from the period of 2006-2015. Left panel illustrates the annual averaged air pollution days in 2006-2015 and right panel shows changes of the pollution days at the end of 21st century with respect to 2006-2015. Dots mean the changes are significant at the 95% confidence level using Student T-test for all years and ensembles. Units: days.

→ [Line 254-257]

Considering the underestimation in aerosol concentration by CESM1 model in China, the percentile threshold metric is also applied here to estimate the future changes in light (90th) and severe (99th) air pollution days. Similar results can be obtained (Fig. S8).

Minor comments:

Line 27: The authors only used one model configuration with fixed anthropogenic emissions. It can definitely be used to rule out the influence of changing emissions. However, for a supplement of the story, I suggest comparing the role of GHG warming and aerosol emission change. I know it is hard to do another simulation, but the authors can easily scale aerosol concentration by emissions using RCP8.5 scenario and roughly predict aerosol concentration under future emissions.

**Reply:** Good suggestion. According to your idea, we have evaluated the future changes of PM$_{2.5}$ concentrations and the associated species along the RCP8.5 forcing trajectory from the large ensemble simulations of CESM1 (Figure 2). Results present different changes of aerosol concentrations when compared with that from the fixed aerosol simulations. More discussion can be found in the following or current manuscript.

[Figure]

**Figure 2.** Simulated linear trends of the total PM$_{2.5}$ surface concentration as well as its associated species (BC, SO$_4$, SOA, and POM) across eastern China for the years of 2006-2099 from the simulations forced by the RCP8.5 trajectory. The linear trends are calculated by the nonparametric Mann-Kendall and Sen's methods, and the significant trends with 0.01 significant level are illustrated by dots. Units: μg/m$^3$/100a.

→[Line 210-219]

For comparison, we also evaluated the future changes of PM$_{2.5}$ concentrations and the associated species along the RCP8.5 forcing trajectory from the large ensemble simulations of CESM1 (Figure not shown). Different from changes of aerosol concentrations under the fixed aerosol simulations, the PM$_{2.5}$ concentrations and the associated species present uniformly decreasing trends across eastern China from the simulations along the RCP8.5 forcing. The decreasing trends in the RCP8.5 simulations are mainly attributed to the prescribed decrease of aerosol forcing in the future in RCP database (Xu and Lin, 2017). The climate change induced by the GHG-warming might exacerbate the air pollution, but the impacts cannot compensate the prescribed decreasing trend of aerosol concentration.

Line 83: The highlight of this study, I guess, it the PM2.5 concentration under global warming. But the PM2.5 change is too small, and the authors showed pollution days instead (68% in abstract), which may not be appropriate and cannot be distinguished to precious studies. The authors may consider re-organize the findings and change the way of presentation.

**Reply:** Thanks for this suggestion and these expressions have been re-organized.

→[Line 30-35]

The anthropogenic PM$_{2.5}$ concentration is estimated to increase averaged over eastern China at the end of this century, but varying from regions, with an increase over northwestern part of eastern China and a decrease over southeastern part. Similar changes can be observed for the light air pollution days. However, the severe air pollution days is reported to increase across eastern China at the end of this century, particularly around the Jing-Jin-Ji region.

Line 125-133: I don't understand what is fraction of attributable risk. The author can give an example and illustrate what is it used for.

**Reply:** The metric of the fraction of attributable risk (FAR) has been widely used for attribute analyses of climate extreme changes. FAR is defined as the 1-P0/P1, where P0 is the probability of exceeding a certain threshold during the reference period and P1 is the probability exceeding the same threshold during a given period. FAR thus presents the quantitative estimations of effects of the specified forcings on the climate changes (Stott et al., 2016). For example, the simulated occurring probability P0 of the specified extreme event in the pre-industrial experiment is estimated to be about 0.0007 compared to a probability P1 in the GHG-forced experiment of about 0.03, which indicates a more than 40-fold increase in probability between the two experiment, implying a FAR of about 0.97. In other words, there are approximately 97% of the extreme increases are mainly attributed to the impact of GHG increases.

Stott, P.M., et al.: Attribution of extreme weather and climate-related events. WIREs Clim Change, 7, 23-41, 2016.

Line 134: The author used stagnation day defined by Horton et al. (2012). It is definition suit to stagnation in China? Is there any previous study used it for stagnation in China? If not, the authors have to evaluate it using historical data of China.

**Reply:** The performance of this stagnation index over China has been evaluated by some early studies. They suggested that this air stagnation definition might not be applicable for China to represent the air pollution condition under the seasonal scales (Feng et al., 2018; Wang et al., 2018). However, the annual mean stagnation generally presents good agreement with that of air pollution across China (Huang et al., 2017; 2018). Here, we mainly focus on the future changes of $PM_{2.5}$ pollution as well as the meteorological conditions from the annual mean perspective. This definition is thus suit to the stagnation analysis in this study. More information can be found in the following or in the current manuscript.

Feng, J., Quan, J., Liao, H., Li, Y., and Zhao, X.: An air stagnation index to qualify extreme haze events in northern China, J. Atmos. Sci., doi:10.1175/JAS-D-17-0354.1, 2018.

Huang, Q., Cai, X., Song, Y., and Zhu, T.: Air stagnation in China (1985-2014): Climatological mean features and trends, Atmos. Chem. Phys., 17, 7793-7805, 2017.

Huang, Q., Cai, X., Wang, J., Song, Y., and Zhu, T.: Climatological study of the boundary-layer air stagnation index for China and its relationship with air pollution, Atmos. Chem. Phys., 18, 7573-7593, 2018.

Wang, X., Dickinson, R., Su, L., Zhou, C., and Wang, K.: PM2.5 pollution in China and how it has been exacerbated by terrain and meteorological conditions, Bull. Amer. Meteorol. Soc., 99(1), 105-119, 2018.

→ [Line 146-152]

Early studies have suggested that this air stagnation definition might not be applicable for China to represent the air pollution condition under the seasonal scales (Feng et al., 2018; Wang et al., 2018). However, the annual mean stagnation generally presents good agreement with that of air pollution across China (Huang et al., 2017; 2018). The changes in the annual mean states of air stagnations over China at the end of 21st century will thus be discussed in the following.

Line 170: Spatial correlation over eastern China?

**Reply:** Yes. It has been corrected.

→[Line 182-184]

A strong spatial correlation (0.69) is found for the annual mean $PM_{2.5}$ concentration between the site observation and median ensemble of CESM1 simulations over eastern China (Fig. S1).

Line 174: I don't agree the bias is primarily due to missing species. The bias of aerosol concentration is more complicated in China, which has been reported in many previous studies (Yang et al., 2017a,b). The causes include uncertainties in aerosol emission amount, emission injection height, course model resolution, lack of nitrate, aerosol treatment in model (e.g., aging processes, chemistry, wet removal)…

**Reply:** Thanks for this valuable comment. This discussion has been corrected and more information can be found in the following or in the current manuscript.

→[Line 186-190]

However, a negative bias is obvious. Early studies (Li et al., 2016; Yang et al., 2017b; c) have documented that this low bias of aerosol concentration simulated by models is much more complicated in China and the causes mainly involve the uncertainties from aerosol emission amount, emission injection height, lack of nitrate, aerosol treatment in model as well as the coarse model resolution.

Line 187: 2% is too small. The authors can focus on different regions and species.

**Reply:** Some expressions in this aspect have been corrected and added according to your suggestion. More information can be found in the following or in the current manuscript.

→[Line 206-209]

Furthermore, the increases of all major $PM_{2.5}$ species in terms of column burden (BC: 11%, $SO_4$: 6%, SOA: 11%, and POM: 11%) show stronger than the surface concentration (BC: 4%, $SO_4$: 2%, SOA: -1%, and POM: 4%).

→ [Line 220-238]

As mentioned above, the PM$_{2.5}$ surface concentration in the two economic zones of YRD and PRD present a negative response to the GHG-induced warming, while the corresponding column burden shows significantly increasing trends (Fig. S3). The decreases of the surface concentration over these two zones are primarily contributed by the changes of SO$_4$ and SOA, while there are no obvious trends for BC and POM (Figs. S4-S7). The robust response of the increased surface wind speed and decreased upper-level wind speed to GHG warming can be partly responsible for the changes of the major PM$_{2.5}$ species in these two zones, which will be further discussed. Over the zones of JJJ and SCB, both the PM$_{2.5}$ concentrations and the associated major PM$_{2.5}$ species present the significantly rising trends throughout the 21$^{st}$ century. For the surface concentration, PM$_{2.5}$ is reported to increase by 3% and 4% in the regions of JJJ and SCB, respectively, at the end of the 21$^{st}$ century. The BC is reported to increase by 4% and 8% for JJJ and SCB, respectively. The other species, such as SO$_4$ and POM, increase by 4% and 4%, respectively, in the JJJ regions and by 2% and 9%, respectively, in SCB regions. Relatively stronger responses can be seen in changes of the column burden for all major species (Figs. S4-S7). The increased concentrations of PM$_{2.5}$ species finally result in significantly increasing trends of the total PM$_{2.5}$ loading over these two regions, which will present a more direct effect on human health.

Line 190: Add SOA in Figure 2.

**Reply:** SOA has been added in Figure 2. Please see more information in PAGE 32 in the current manuscript.

Line 200: Do the future changes in meteorology (winds, precipitation) over China also exist in other models? At least add literatures.

**Reply:** Yes, similar changes can be found in the other global climate models and regional climate models. More discussions have been added in the text.

→[Line 296-300]

The decreasing trend of wind speed in the 21$^{st}$ century across China not only exists in CEMS1 model, but also happens in the other global climate models that participated in Coupled Model Intercomparison Project Phase 3 (CMIP3) and CMIP5 (Jiang et al., 2010a; Mclnnes et al., 2011), as well as in regional climate models (Jiang et al., 2010b).

→[Line 325-328]

The future changes of precipitation days present much robust. Both the increasing trends of heavy precipitation days and the decreasing trends of light precipitation days are also obvious across China simulated by the CMIP5 models (Chen and Sun, 2013; 2018), as well as the regional climate models (Gao et al., 2012).

Line 217: I don't think it is a good idea to emphasize and severe days and use 'robust response'. First, PM2.5 > 75 is suit for observations. The simulated PM2.5 is only 1/3 of the observation value. Second, I don't think the change will large than the standard deviation of severe days for different ensemble simulations and years. BTW, do you mean 'positive' response.

**Reply:** Thanks for this valuable comment and this expression has been re-organized.

→[Line 248-252]

In contrast to the light air pollution days, the severe anthropogenic air pollution days (PM$_{2.5}$ > 75 μg/m$^3$) show a positive response to the GHG-induced warming across eastern China, particularly for the regions around JJJ in which the high PM$_{2.5}$ concentration was localized (Fig. 3).

Line 226: 'PM2.5 loadings and their associated pollution days still present significant increases'. As mentioned above, I don't think this statement is correct.

**Reply:** It has been corrected.

→[Line 259-262]

Although the aerosol emission was constant throughout the experiment, our study reveals that the PM$_{2.5}$ loadings and their associated pollution days still present increases throughout the 21$^{st}$ century, primarily resulting from the impact of climate change induced by GHG warming.

Line 228-246: I don't understand what this section is used for. '28% of the pollution days are contributed by the climate change that was induced by GHG warming.' The authors fixed aerosol emission and all changes (100%) in the model should be due to GHG warming. Even changes in pollution days due to changes in meteorological conditions result from GHG warming.

**Reply:** In this section, we applied the "Fraction of Attributable Risk (FAR)" metric to roughly estimate the possible contribution of climate change impact on the air pollution. The metric FAR has been widely used for the attribute analyses of climate extreme changes and it can present the quantitative estimations of effects of the GHG-induced climate changes on the anthropogenic air pollutions (More information referring to the calculation can be found in the text). The experiments are used in this study with the fixed aerosol emission and the air pollution changes are mainly contributed by the climate change that induced by GHG warming, which can be estimated by the FAR metric.

Line 293-295: Again, 'substantially increase' is not correct.
**Reply:** It has been corrected.

→[Line 333-336]

[revised manuscript text omitted]

---

## Author Comment (AC2) · 12 Sep 2018

**Response to Reviewer #2:**

Chen et al. attempted to elucidate how PM pollution in eastern China will response to future GHG warming, using a large ensemble of CESM simulations. The authors reported that GHG-induced climate change will increase PM pollution days, especially the most severe polluted days (PM2.5>75 µg m-3), at the end of 21th century and they argued that reduced tropospheric winds and light precipitation days can be the reasons. Their results are interesting and could deepen our understanding of the impacts of climate change on air quality. The topic is suitable for ACP readers, and this paper is well structured. However, I have some concerns about the linkage between pollution increase and changes in meteorology. The authors need to address the following comments before it can be published.

**Reply:** Greatly thanks for your valuable comments and suggestion, which have been fully considered and corrected in the current manuscript.

General Comments:

- The authors found an increase of 68% in the most severe pollution days, with only an increase of 3% for light pollution days, but they attributed such increase to the mean change of future GHG-induced climate change. In statistics, I think the increase in most severe pollution days represents the extreme cases, whose linkage to mean climate change needs to be further explored, or at least discussed.

**Reply:** Thanks for this comment. Yes, the severe pollution events are the extreme cases that largely associated with extreme anomalies of the meteorological conditions. However, it is a great challenge to discuss the changes of the associated meteorological factors for each extreme case, due to the low capability of current model simulation in extremes. In this study, we mainly focus on the impact of mean climate changes on the air pollutions. As we all know, if the mean climate state moves to a level that not favorable for the air pollutant dissipation, the change of the mean meteorological condition would increase the occurring probability of the extreme pollution days. This is the main view we hope to show in this study.

According to the suggestions from you and another reviewer, we have re-organized some expressions referring to the changes of pollution days. More information can be found in the following or in the manuscript.

→[Line 239-257]

The increase in $PM_{2.5}$ surface concentration throughout the 21st century substantially leads to the significant increase of the light anthropogenic $PM_{2.5}$ pollution days ($PM_{2.5} > 25 \mu g/m^3$) across the northwestern part of eastern China (Fig. 3). Due to the decrease of $PM_{2.5}$ concentration over the southeastern part of eastern China, the light anthropogenic air pollution days can be expected to decrease in this region. Estimation shows that the number of the light air pollution days would be decreased by approximately 10 days at the end of the 21st century with respect to the early period of this century in the region. However, the annual mean light air pollution days is reported to increase averaged over the eastern China at the end of this century despite the aerosol emission is constant throughout the experiment. In contrast to the light air pollution days, the severe anthropogenic air pollution days ($PM_{2.5} > 75 \mu g/m^3$) show a positive response to the GHG-induced warming across eastern China, particularly for the regions around JJJ in which the high $PM_{2.5}$ concentration was localized (Fig. 3). The severe air pollution days is estimated to increase by more than 2 days at the end of this century when compared to the early period over this region. Considering the underestimation in aerosol concentration by CESM1 model in China, the percentile threshold metric is also applied here to estimate the future changes in light (90th) and severe (99th) air pollution days. Similar results can be obtained (Fig. S8).

- The ACCMIP (Lamarque et al., 2013) also archives similar simulations by several climate models. It would be helpful if the authors can compare their results with ACCMIP models. Just a suggestion.

**Reply:** Thanks for your recommendation. We have read this paper carefully and found some interesting and valuable information, which has substantially improved our knowledge. However, we have not added the comparison discussions of the ACCMIP simulations with the CESM1 results, mainly due to the three reasons. First, the analyses in this study are implemented mainly from the global scale, which shows weak comparison with our analyses that just limited in China. Second, the ACCMIP simulations are forced along the RCP8.5 emissions, but the aerosol emissions are fixed at the current level in CESM1 simulations. The comparison may be not much robust due to the different model and different forcing. Third, we have added the comparison discussion between the simulations that forced along the RCP8.5 trajectory and the fixed aerosol emission from CESM1 model.

→ [Line 210-219]

For comparison, we also evaluated the future changes of $PM_{2.5}$ concentrations and the associated species along the RCP8.5 forcing trajectory from the large ensemble simulations of CESM1 (Figure not shown). Different from changes of aerosol concentrations under the fixed aerosol simulations, the $PM_{2.5}$ concentrations and the associated species present uniformly decreasing trends across eastern China from the simulations along the RCP8.5 forcing. The decreasing trends in the RCP8.5 simulations are mainly attributed to the prescribed decrease of aerosol forcing in the future in RCP database (Xu and Lin, 2017). The climate change induced by the GHG-warming might exacerbate the air pollution, but the impacts cannot compensate the prescribed decreasing trend of aerosol concentration.

Specific Comments:

-Line 32-34: As indicated above, the authors should take care here.

**Reply:** Thanks for this suggestion. We have toned down this expression.

→ [Line 36-38]

Further research indicates that the increased stagnation days and the decreased light precipitation days are the possible causes of the increase in $PM_{2.5}$ concentration, as well as the anthropogenic air pollution days.

-Line 134-140: The relationship between air stagnation index used here and PM2.5 pollution in China may be not well correlated (e.g., Feng et al., 2018).

**Reply:** The performance of this stagnation index over China has been evaluated by some early studies. They suggested that this air stagnation definition might not be applicable for China to represent the air pollution condition under the seasonal scales (Feng et al., 2018; Wang et al., 2018). However, the annual mean stagnation generally presents good agreement with that of air pollution across China (Huang et al., 2017; 2018). Here, we mainly focus on the future changes of PM$_{2.5}$ pollution as well as the meteorological conditions from the annual mean perspective. This definition is thus suit to the stagnation analysis in this study. More information can be found in the following or in the current manuscript.

Feng, J., Quan, J., Liao, H., Li, Y., and Zhao, X.: An air stagnation index to qualify extreme haze events in northern China, J. Atmos. Sci., doi:10.1175/JAS-D-17-0354.1, 2018.

Huang, Q., Cai, X., Song, Y., and Zhu, T.: Air stagnation in China (1985-2014): Climatological mean features and trends, Atmos. Chem. Phys., 17, 7793-7805, 2017.

Huang, Q., Cai, X., Wang, J., Song, Y., and Zhu, T.: Climatological study of the boundary-layer air stagnation index for China and its relationship with air pollution, Atmos. Chem. Phys., 18, 7573-7593, 2018.

Wang, X., Dickinson, R., Su, L., Zhou, C., and Wang, K.: PM2.5 pollution in China and how it has been exacerbated by terrain and meteorological conditions, Bull. Amer. Meteorol. Soc., 99(1), 105-119, 2018.

→[Line 146-152]

Early studies have suggested that this air stagnation definition might not be applicable for China to represent the air pollution condition under the seasonal scales (Feng et al., 2018; Wang et al., 2018). However, the annual mean stagnation generally presents good agreement with that of air pollution across China (Huang et al., 2017; 2018).

The changes in the annual mean states of air stagnations over China at the end of 21st century will thus be discussed in the following.

-Line 148 and Figure 1: Why chose a reference concentration of 75 µg m-3. The annual PM2.5 standard in China is 35 µg m-3.

**Reply:** This reference concentration of 75 µg m$^{-3}$ is selected in this study according to the guideline value from the World Health Organization (2014). The 24-hour concentration of PM$_{2.5}$ exceeding 75 µg m$^{-3}$ is considered as the standard value of Interim target-1 that presented a high possibility of exposing people to serious health hazards. This value is determined basing on published risk coefficients from multi-centre studies and meta-analyses. Some related discussions have been available in the manuscript.

World Health Organization: Air quality guidelines: Global update 2005. World Health Organization Rep., 496 pp., www.euro.who.int/_data/assets/pdf_file/0005/78638/E90038.pdf/, 2014.

→[Line 160-165]

According to the statistics, there are approximately 95% sites where the annual mean PM$_{2.5}$ concentration exceeded the WHO recommended 24-hour standard (25 µg/m$^3$) across eastern China, and there are 65 sites centralized by Beijing, where the annual mean PM$_{2.5}$ concentration was larger than 75 µg/m$^3$, which would present the possibility of exposing people to serious health hazards (World Health Organization, 2014).

-Line 170-172: The correlation is based on what observational and model data. Should make it clear.

**Reply:** This spatial correlation is calculated between the site observation and median ensemble of CESM1 simulations.

→[Line 182-184]

A strong spatial correlation (0.69) is found for the annual mean $PM_{2.5}$ concentration between the site observation and median ensemble of CESM1 simulations over eastern China (Fig. S1).

-Line 173-175: Same as above, the low bias in model depends on what observational PM2.5 you used. As reported in Li et al. (2016), the RCP emissions for year 2005 underestimated anthropogenic emissions of aerosol precursors over China. Thus, the lower PM2.5 concentration in model could also partly attribute to underestimated emissions.

**Reply:** Thanks for this valuable comment. This discussion has been corrected and more information can be found in the following or in the current manuscript.

→[Line 186-190]

However, a negative bias is obvious. Early studies (Li et al., 2016; Yang et al., 2017b; c) have documented that this low bias of aerosol concentration simulated by models is much more complicated in China and the causes mainly involve the uncertainties from aerosol emission amount, emission injection height, lack of nitrate, aerosol treatment in model as well as the coarse model resolution.

-Line 182-183: Which region you average the PM2.5 concentration for "eastern China"?

**Reply:** In this study, the region of east to 100 °E is considered as the eastern China, which has been indicated in the manuscript.

→[Line 101-103]

In our study region of eastern China (east to 100 °E), there are 1263 sites remaining after the sites with missing values were removed during 2015-2017.

-Line 205: What "SC" shorts for?

**Reply:** Thanks for this comment. "SC" in the early manuscript is short for "Sichuan Basin", which has been corrected to "SCB" in the current manuscript. It has been corrected through the manuscript and all figures.

[revised manuscript text omitted]